# Morphology and Growth Mechanism of β-Rhombohedral Boron and Pentagonal Twins in Cu Alloy

**Junqing Han** [1,†], **Wentao Yuan** [1,†], **Yihan Wen** [1], **Zuoshan Wei** [2], **Tong Gao** [1], **Yuying Wu** [1,*] and **Xiangfa Liu** [1]

1   Key Laboratory of Liquid-Solid Structure Evolution and Processing of Materials, Ministry of Education, Shandong University, Jinan 250061, China
2   Shandong Key Laboratory of Advanced Aluminum Materials and Technology, Binzhou Institute of Technology, Binzhou 256600, China
*   Correspondence: wuyuying@sdu.edu.cn
†   These authors contributed equally to this work.

**Abstract:** In this work, boron particles with β-rhombohedral structure were prepared in Cu-4B alloy. The morphology and growth mechanism of β-B and pentagonal twins were analyzed. Results show that boron crystals possessed an approximate octahedral structure which consisted of two planes belonging to {001} facet and a rhombohedron formed by {101} planes. The morphology of the boron crystal was determined by the position and size of {001} planes. During growth, parts of boron crystal formed twins to reduce surface energy. Five particular single crystals can shape a pentagonal twin. The morphological distinction between pentagonal twins mainly came from the difference in morphology of single crystal. When the {001} exposed planes were large and showed a hexagonal shape, the boron crystal often formed parallel groupings and polysynthetic twins to reduce surface energy.

**Keywords:** β-rhombohedral boron; twins; pentagonal twin; growth mechanism

## 1. Introduction

In the aviation industry, boron (B) is considered the most ideal combustion aid for jet fuel, due to having the highest volumetric heating value and extremely high gravimetric heating, which is second only to that of beryllium (Be) [1]. 10B, isotope of boron, possesses a relatively strong neutron absorbing capacity [2], so 10B and its composite materials are widely used for boron neutron capture therapy [3] and thermal neutron detectors [4]. Many different shapes of boron have been successfully prepared by chemical and physical methods, such as boron nanotubes, [5,6] boron nanowires, [7,8] boron nanocones, [9] and boron nanoribbons. [10,11] In addition, in our previous studies, we also spheroidized eutectic boron by varying the cooling rate and adding alloying elements to prepare submicron boron spheres and hollow boron spheres. [12] However, the morphology and structure of boron have a great influence on its application. For instance, Evgeni S. Penev discovered that boron had the potential to transform a superconductor when the boron possessed a two-dimensional structure [13].

Allotropy of boron has been widely reported in recent years, as seen with γ-B [14] and t-B [15], but the growth model and mechanism of boron crystal are still focused on the field of simulation. Wataru Hayami calculated the surface energy of α-B [16] and t-B [17], and gave the crystallographic monomorph of these two allotropies. However, the structures of t-B and γ-B are different from those of rhombohedral boron. Under normal temperature and pressure, boron often exists in the form β-B (a = 10.145 ± 0.015 Å, α = 65°17′ ± 8′) [18] and each lattice point is occupied by an icosahedron (B12) [19,20]. Both α-B and β-B have a rhombohedral structure, the space group of $R\bar{3}m$ (group no. 166). The B12 in α-B is distributed at each lattice point of the rhombohedral unit cell, while the B12 of β-B is not only distributed at the lattice points, but also at the center of the edge of the rhombohedral

unit cell. In addition, there are two triple-fused B28 polyhedrons in the center of β-B unit cell, and these two polyhedrons are linked by a gap boron atom [20,21]. In an experiment of using aluminum-doped β-rhombohedral boron, a three-dimensional framework made of $B_{12}$ icosahedra with voids being occupied by the $B_{28}$–B–$B_{28}$ units was found [22]. Sun [23] et al. studied the growth mechanisms of alpha-boron (α-B) and beta-boron (β-B) in Cu-B alloys in copper melts and observed lamellar growth traces and twin structures of alpha-boron (α-B) and beta-boron (β-B) and produced a model for the growth of β-B, but there is little information about their monotype and exposed surface.

Because boron has a special three-center bond structure [24,25], the dislocations are difficult to form. To reduce the surface energy and release stress, boron crystals often twin during crystallization [23] and ball milling [26]. Some particular pentagonal twin structures have been found in $B_4C$ [27] and gold nanocrystal [28], but there are almost no reports for boron. In this work, the morphology of β-B and pentagonal twins were reported and analyzed.

## 2. Experimental Section

Pure Cu (>99.7 wt%) and pure B (99.9 wt%) were used to prepare Cu-4B (with the same weight percentage as that reported below, unless otherwise specified) alloy. Pure Cu was melted using a high-frequency stove. After Cu melt, the B enfolded by copper foil was added to the melt. Then, the melt was poured into a cast-iron mold to obtain Cu-4B alloy. Small alloy blocks cut from Cu-4B alloy were re-melted by a high-frequency induction coil to obtain blow-cast alloys and rapidly solidifying alloy strips. During the preparation of rapidly solidifying alloy, the rotating speeds of the rotated copper mold with a perimeter of 690 mm were 1500 r/min and 3000 r/min. The cooling of Cu-4B alloy ingots was 100 K/s, and the cooling rate of alloy ingots, blow-cast alloys and alloy strips increased in turn.

The Cu in Cu-4B alloy ingots, blow-cast alloys and alloy strips was eroded by 50% $HNO_3$, and the B powders remained. The B powders were repeatedly cleaned with deionized water until the PH reached 7. The micro-morphology of boron was characterized by field emission scanning electron microscope (JSM-7800F SEM, Japan). The chemical composition of boron was analyzed using the JEM-2100F high-resolution transmission electron microscope linked with an energy dispersive X-ray spectroscopy (EDS) attachment and Oxford XMax80 spectrometer (SU-70, Japan). The crystal structure was analyzed using a transmission electron microscope (TEM, JEM-2100F, Japan).

## 3. Results and Discussion

The morphology of Cu-4B alloy and boron particles extracted from Cu-4B ingots, blow-cast alloys and alloy strips (1500 r/min and 3000 r/min) is shown in Figure 1. The white arrows in Figure 1a indicate some special structures, such as hexagonal boron, twins and pentagonal twins. The particles shown in Figure 1b–f were typical primary boron [23]. As shown in Figure 1b, the diameter of boron particles was about 5–10 μm. The EDS result in Figure 1c confirms that these crystals were indeed boron, and the existence of the concentration of Au was caused by the gold spray treatment before the SEM test. As shown in Figure 1d, compared with Figure 1b, the diameter of boron particles extracted from blow casting alloy was relatively small. The size of boron particles extracted from alloy strips was about 3–5 μm, no matter what rotation rate was used. As for the Cu-B binary phase diagram [29], the primary boron in Cu-4B wt% (about Cu-20B at%) alloy was precipitated at 1050 °C, while the eutectic boron was precipitated at 1013 °C, and the cooling rate was relatively high; therefore, the time needed for crystals to grow was very short. Both in blow-cast alloy and alloy strips, the crystals did not grow enough, so the size of particles extracted from blow casting alloy and alloy strips was similar. Although the size of the boron particles extracted from ingots alloys or strips was slightly different, there was almost no distinction in terms of morphology. This indicates that in this work, the cooling rate did not affect the crystal structure of boron. Therefore, the source of boron crystal was not considered in the following crystal morphology analysis.

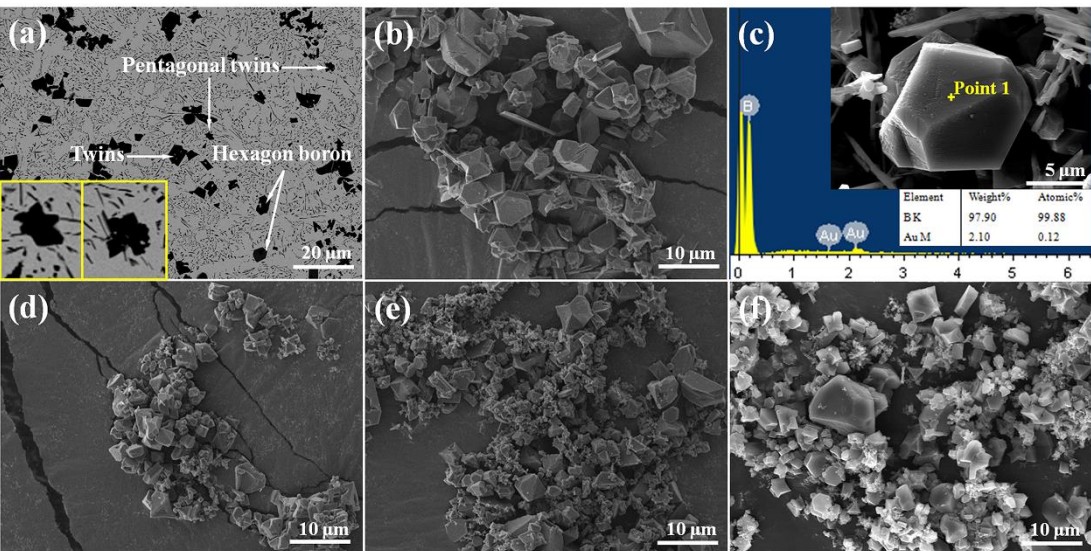

**Figure 1.** (**a**) Backscattered electron (BE) image of Cu-4B alloy and the enlarged images of the boron particles with pentagonal twins. Secondary electron (SE) images of boron particles extracted from (**b**) alloy ingots, (**c**) EDS result of boron crystal extracted from alloy ingots, (**d**) blow-cast alloy, (**e**) alloy strips (1500 r/min) and (**f**) alloy strips (3000 r/min).

Figure 2 shows several typical crystal morphologies of boron, with partial planes and facet families were marked in Figure 2a–f. According to the calculation results of Hayami et al. [16], the {001} and {101} have lower surface energy, and the (001) facets have the lowest surface energy for the trigonal structure β-B. The exposed surfaces of these boron crystals were {001} and {101} planes. These crystals can be considered as a structure formed by a completed rhombohedron composed of {101} planes and cut by (001) plane. The morphological difference of these boron crystals is mainly due to the location of {001} planes. As shown in Figure 2a, when the (001) plane closed to the top of the crystal, the shape of the (001) plane was a triangle and the {101} planes were pentagons. As the (001) plane descended, the size of the (001) plane was gradually increased. When the (001) plane dropped to the position shown in Figure 2b, the size of the triangular (001) plane reached its maximum. At this moment, the morphology of the boron crystal was similar to that of an octahedron [30]. Together with {001} planes, all {101} planes were triangles. When the (001) plane was in the middle of the boron crystal, the shape of {001} and {101} planes transformed into hexagons and trapezoids, respectively, as shown in Figure 2c,f. In order to further confirm the crystal structure of boron crystal, TEM and selected electron diffraction were used. As shown in Figure 2g, the morphology of boron crystals was mainly hexagonal under TEM. A relatively thin hexagonal boron crystal, as shown in Figure 2h, was chosen for selected electron diffraction. The diffraction pattern confirmed that the boron particle possessed β-B structure, and the diffraction spots correspond to the (201), (306) and (105) planes of β-B. The results of selected area electron diffraction are consistent with those of Sun et al. [23], suggesting that the pentagonal B particles also have the structure of β-B.

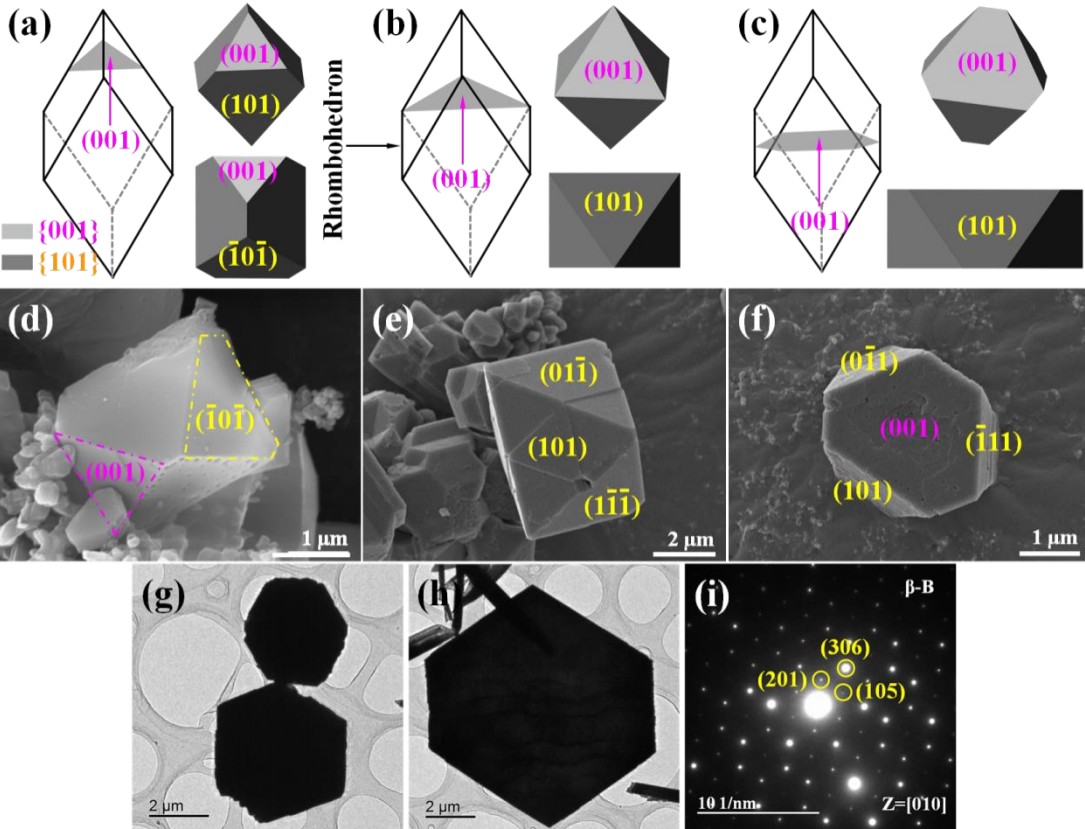

**Figure 2.** (**a**–**c**) Schematic diagrams of three typical crystal morphologies of boron crystal; (**d**–**f**) SEM images of boron crystals correspond to Figure 2a–c, respectively. (**g**,**h**) TEM images of boron crystals extracted from Cu-4B alloy ingot. (**i**) Diffraction pattern of boron crystal in Figure 2h. All the exposed surfaces of completed rhombohedron in (**a**–**c**) were {101} planes.

During the growth of boron crystal, twins are very common [31]. Figure 3 shows some of the twin structures of boron. The pentagonal twin in Figure 3a was shaped by complete rhombohedral boron crystals. To form a pentagonal structure, angle 1 must be close to 72°. The (101) plane is marked in Figure 3a. Another pentagonal twin, which was formed by the crystals shown in Figure 2a,b, can be seen in Figure 3b. Like the pentagonal twin shown in Figure 3a, the theoretical value of angle 2 was also close to 72°. The pentagonal twin was not perfect; there were grooves and protrusions in the twin, as shown by the white arrows. A side view of the pentagonal twin is shown in Figure 3c, and the crystals that formed the pentagonal twin are marked by yellow and green lines, respectively. The theoretical value of the plane angle between $(01\bar{1})$ and $(\bar{1}11)$ was 72.8°, which was closed to angles 1 and 2. This pentagonal twin crystal has morphological similarity with the penta-twinned gold nanocrystals discovered by Zhang et al. [32] As shown in Figure 3d, when the {001} planes showed the hexagonal shape in Figure 2c, it was difficult for boron crystals to form pentagonal twins. This type of boron crystal is prone to take shape original twins, parallel grouping, as well as the polysynthetic twin that can be observed in the white box in Figure 3c.

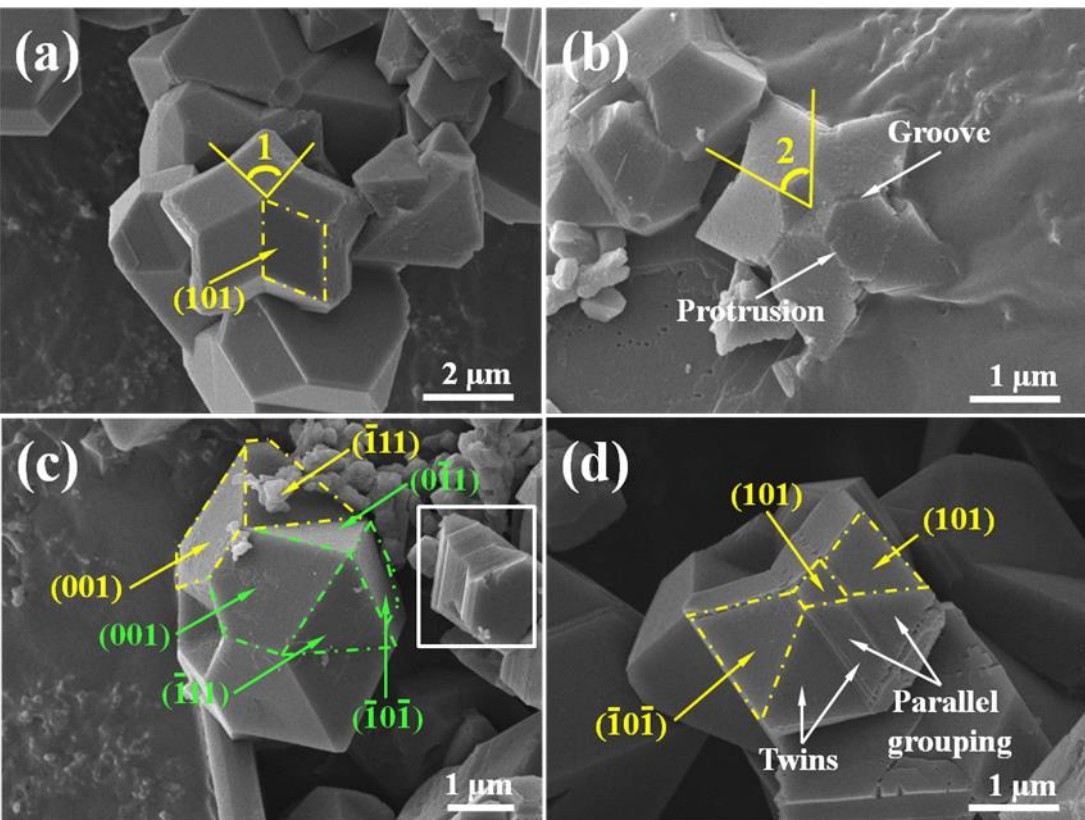

**Figure 3.** (**a**,**b**) Pentagonal twins, (**c**) the side view of pentagonal twin, (**d**) twins and parallel grouping. Parts of planes belonging to {101} and {001} are marked in Figure 3.

Figure 4 is the schematic diagram of the growth of several twins. Different morphological crystals can form distinct twins. As shown in Figure 4a, the original rhombohedral boron crystals could form the pentagonal twin I shown in Figure 3a without being cut by {001} planes. First, rhombohedral boron crystal was twinned, and the twin plane was $(01\bar{1})$ exposed planes. Then, the crystal 2, 3 and 5 emerged by twinning, and their twin planes were $(01\bar{1})$ or $(1\bar{1}\bar{1})$ exposed face. Finally, the pentagonal twin was finished. The formation process of pentagonal twin II shown in Figure 4b was similar to that of pentagonal twin I, both adopting $(01\bar{1})$ or $(1\bar{1}\bar{1})$ planes as the twin plane. Most of its monomers are the crystals shown in Figure 2e (simple form 2), that is, octahedral like crystals, and sometimes the crystals shown in Figure 2d also exist. For example, the pentagonal twins in Figure 3b were all composed of octahedral like crystals, while the pentagonal twin II in Figure 3c had the crystals shown in Figure 2d, which are marked with yellow dotted lines. However, the twin plane of pentagonal twin II was located inside crystal 1. As shown in Figure 4b, when the twin plane is inside, the edges of crystal 1 and crystal 2 will intersect. Therefore, after twinning, two new edges were formed on (001) and (101) planes. Edge 1 corresponded to the groove shown in Figure 3b and edge 2 participated to shape angle 2 in Figure 3b. After the formation of rhombohedral boron crystals 3–5, pentagonal twin II with a sunken pentagon in the center formed. The five new edge 2s and the original edges of the crystal together formed the pentagonal funnel shaped defect in Figure 3b.

Boron crystals that have hexagonal {001} planes can take the shape of common twins, parallel groupings, and polysynthetic twins. As shown in Figure 4c, boron crystals like simple form 3 can take the (001) plane as the twin plane to form twin. The twin can continue to grow in this shape, or can turn into a parallel grouping and polysynthetic twin, as shown in the white box in Figure 3c. If single crystal 1 forms a pentagonal twin, the large {001} planes will be exposed, and there will be a hole in the middle of this pentagonal twin. These two phenomena are both harmful in reducing the overall surface energy. On the contrary,

the formation of twins as shown in Figure 4c by sharing {001} planes can significantly reduce the surface energy.

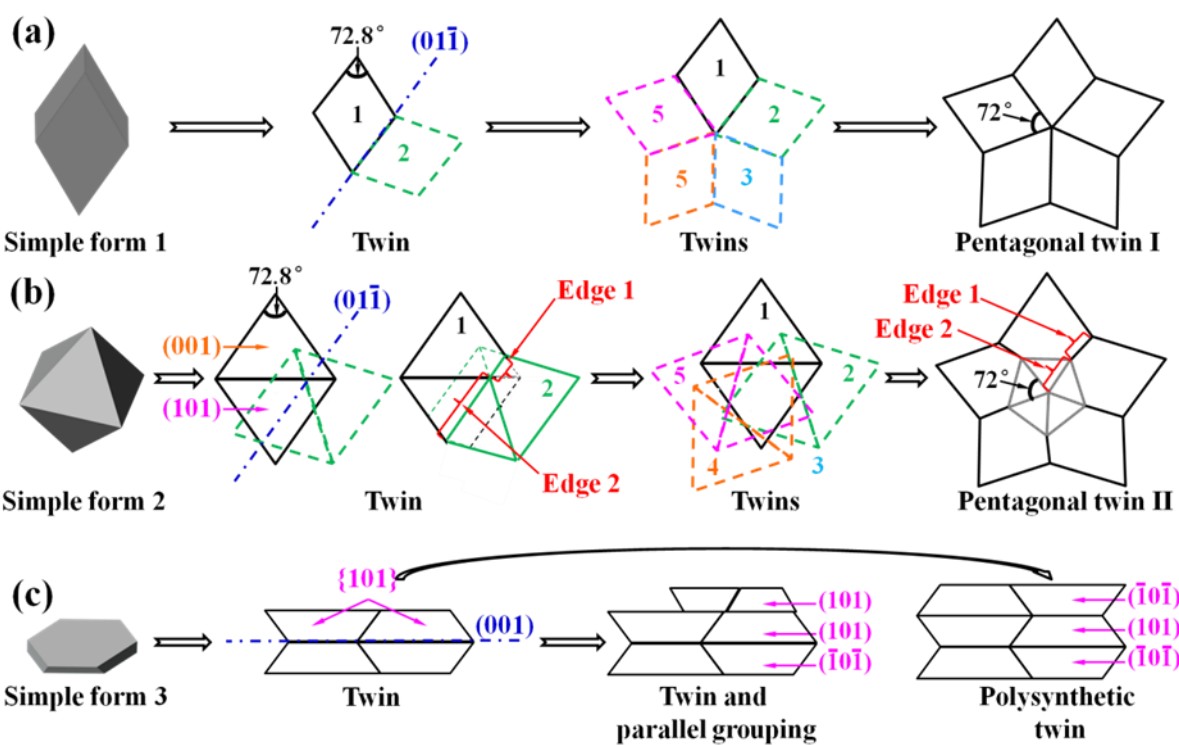

**Figure 4.** Schematic diagram of twin growth of (**a**) pentagonal twin I, (**b**) pentagonal twin II and (**c**) twins, parallel grouping, and polysynthetic twin.

## 4. Conclusions

In the Cu-4B alloy, boron particles mainly possessed the β-B structure. Three typical boron crystal morphologies were analyzed and their exposed surfaces were {001} and {101} planes. The morphological difference between them was due to the position of {001}. Boron was prone to twinning during growth, and the complete rhombohedron formed by {101} planes could form relatively complete pentagonal twins. Boron crystals enclosed by triangular {001} planes and {101} planes could take the shape of pentagonal twins with a sunken pentagon in the center. When the {001} planes transformed into hexagons, it was easy for boron crystals to form parallel grouping and polysynthetic twin to reduce surface energy.

**Author Contributions:** Conceptualization, J.H. and W.Y.; methodology, W.Y.; software, W.Y. and J.H; formal analysis, W.Y. and J.H.; writing—original draft preparation, W.Y., J.H., Y.W. (Yuying Wu), Y.W. (Yihan Wen) and X.L.; writing—review and editing, J.H., W.Y., Y.W. (Yuying Wu), Z.W., T.G., Y.W. (Yihan Wen) and X.L.; visualization, J.H., W.Y. and Z.W.; supervision, Y.W. (Yuying Wu); project administration, Y.W. (Yuying Wu); funding acquisition, Y.W. (Yuying Wu) All authors have read and agreed to the published version of the manuscript.

**Funding:** The National Key R&D Program of China (2021YFB3400800), the Key Research and Development Program of Shandong Province (Grant No. 2021ZLGX01 and 2021SFGC1001), and the Shandong University Climbing Program Innovation team.

**Institutional Review Board Statement:** Not applicable.

**Informed Consent Statement:** Not applicable.

**Data Availability Statement:** The data presented in this study are available in article.



**Acknowledgments:** This study was financially supported by the National Key R&D Program of China (2021YFB3400800), the Key Research and Development Program of Shandong Province (Grant No. 2021ZLGX01 and 2021SFGC1001), and the Shandong University Climbing Program Innovation team.

**Conflicts of Interest:** The authors declare no conflict of interest.

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
