# Peer review of "Morphology and Growth Mechanism of β-Rhombohedral Boron and Pentagonal Twins in Cu Alloy"

_crystals, doi:10.3390/cryst12111516_

Round 1

Reviewer 1 Report

The paper should be published. However, I would feel more comfortable as a future reader of the published paper if the authors could improve the introduction by making a reference to any previous work done on the topic of assisted growth of beta-B.

Reviewer 2 Report

1.     The chemistry composition of boron was analyzed by an energy dispersive spectroscopy (EDS) detector (JEM-2100F, Japan) (lines 66-68).

JEM-2100F is a transmission electron microscope, not an energy dispersive spectroscopy (EDS) detector.

2.     Backscattered electron (BE) image of Cu-4B alloy and the enlarged image of two pentagonal twins (lines 91-92).

These enlarged images are not images of twins, but images of the boron particles with pentagonal twins.

3.      The white arrows in Figure 1a indicated some special structures, such as hexagon boron, twins, and pentagonal twins.

Why these structures are special? These are just boron particles with or without twins. What does it means hexagon boron? Morphology of boron particle?

4.     The size of boron particles extracted from alloy strips was about 3-5 μm, no matter what rotation rate was used.

These data was obtained from SEM study. TEM could demonstrate other size distribution. More TEM investigations of the boron particles are necessary.

5.     The pentagonal twin was not perfect, there were groove and protrusion in the twin.

The TEM study of twin boundaries could clarify the issue. Surely, FIB/TEM study will be of benefit, but may be will presented in next paper.

6.     Does the pentagonal twinning occur during pure boron crystal growth (not in the Cu alloy)? 

Reviewer 3 Report

 "Results and discussion"  should be  strengthened to other information and data, Fig. 1-4 is similar to information,  lack to the other topic.

Reviewer 4 Report

1. Out of the 20 references, the first 19 are in the introduction section. Since this paper does not corroborate its findings or discussions with the earlier works, it is difficult to forecast the reproducibility of the results. It is suggested that the findings and hypotheses proposed in the results and discussions section be justified in accordance to the theory or observations of related works.

2. How is figure 4, the twin growth (mechanism) schematic diagram? First of all, it is proposed without any association of spatial or temporal phenomena associated with the experimental observations or kinetic theory of crystal growth. It just looks like a geometric sketch of the twin morphologies. In summary, Figure 5 must be associated with the observations mentioned in earlier figures.

Round 2

Reviewer 4 Report

I recommend for the acceptance of the manuscript.